# Dependence of Incidence Angle and Flux Density in the Damage Effect of Atomic Oxygen on Kapton Film

**DOI:** 10.3390/polym14245444

**Published:** 2022-12-12

**Authors:** Wang Zhao, Qiang Wei, Chuanjin Huang, Yaoshun Zhu, Ning Hu

**Affiliations:** 1Key Laboratory of Hebei Province on Scale-Span Intelligent Equipment Technology, School of Mechanical Engineering, Hebei University of Technology, Tianjin 300401, China; 2State Key Laboratory of Reliability and Intelligence Electrical Equipment, Hebei University of Technology, Tianjin 300401, China

**Keywords:** Kapton film, atomic oxygen, incidence angle, flux density, ReaxFF, molecular dynamics simulation

## Abstract

Kapton film is a polymeric material widely used on low-Earth-orbit (LEO) spacecraft surfaces. In the LEO environment, atomic oxygen (AO) is spaceflight materials’ most destructive environmental factor. The erosion mechanism of AO on Kapton films has long been an important issue, where the parameter dependence of the AO effect has received increasing attention. Studies of AO energy and cumulative flux have been extensively carried out, while the influence mechanism of the incidence angle and flux density is not fully understood. The AO incidence angle and flux density in space are diverse, which may cause different damage effects on aerospace materials. In this paper, the dependence of the incidence angle and flux density in the damaging effect of AO on Kapton films was investigated using ground-based AO test technology and the reactive molecular dynamics (ReaxFF MD) simulation technique. Firstly, the ground-based experiment obtained the mass loss data of Kapton films under the action of AO with a variable incidence angle and flux density. Then, the mass loss, temperature rise, product, and erosion yield of Kapton during AO impact with different incidence angles and dose rates were calculated using the ReaxFF MD method. The influences of the incidence angle and flux density on the damage mechanism of the AO effect were discussed by comparing the simulation and test results. The results show that the AO effect in the lower incidence angle range (0–60°) is independent of the incidence angle and depends only on the amount of impacted atomic oxygen. AO in the higher incidence angle range (60–90°) has a surface stripping effect, which causes more significant mass loss and a temperature rise while stripping raised macromolecules from rough surfaces, and the erosion effect increases with the increasing incidence angle and amount of impacted atomic oxygen. There is a critical value for the influence of flux density on the AO effect. Above this critical value, AO has a reduced erosive capacity due to a lower chance of participating in the reaction. The amount of each main product from the AO effect varies with the incidence angle and flux density. Nonetheless, the total content of the main products is essentially constant, around 70%. This work will contribute to our understanding of the incidence angle and flux density dependence of the AO effect and provide valuable information for the development of standards for ground simulation tests.

## 1. Introduction

Kapton is a polyimide (PI) film. This polymer material is widely used in structural, thermal control, and thermal insulation components of low-Earth-orbit (LEO) spacecraft due to its excellent thermal stability, electrical insulation, lightweight flexibility, and UV stability [1]. However, environmental factors such as atomic oxygen (AO), space debris, energetic particles, high and low temperatures, and solar radiation in the LEO environment can affect the performance of Kapton films and pose a potential hazard to the reliable operation of spacecraft [2]. AO, formed by oxygen molecules under the photolysis of solar radiation, accounts for 80% of the atmospheric component of the LEO environment and is considered the most destructive environmental factor in LEO [3,4].

The erosion mechanism of the AO effect on spaceflight materials has been widely studied. On the one hand, AO is highly oxidizing and can react directly with materials chemically; on the other hand, AO in the LEO environment is characterized by high flux and high energy (4–5 eV), which can cause physical stripping and performance degradation in materials when it impacts spacecraft surfaces [5,6,7,8,9,10]. However, the parameter dependence of the AO effect has been less studied. Experimental and simulation studies have only been conducted on the influences of the AO energy and cumulative flux, while the influence mechanism of incidence angle and flux density is not fully understood [11,12,13,14,15,16]. Recently, the impact of the AO energy and dose rate on the damaging effect of PI and PI/POSS composites was investigated using a reactive molecular dynamics (ReaxFF MD) simulation method, where the dose rate had the same function as the flux density in expressing the AO impact frequency [16]. The results indicated that the synergistic effect of high energy and a high dose rate significantly accelerated the erosion of the material. However, the focus of this study was not on the parameters of AO, and the question of whether increasing the dose rate changed the mechanism of AO action was not explored in depth. In terms of incidence angle studies, limited data from early airborne flight tests suggested that Kapton’s mass loss rate was related to the incidence angle of AO in accordance with cos^1.5^θ [17,18]. However, ground-based tests using a laser atomic oxygen system reported Kapton’s mass loss data independent of the incidence angle. The authors concluded that the rough surface of the polymer and the multiple-bounce effect counteracted the effect of the incidence angle [19]. The reasons for the inconsistent results of incidence angle dependence studies from flight and ground tests remain uncertain. Conventional experiments combined with material characterization present difficulties in studying the erosion mechanisms of AO at different incidence angles and flux densities. Other research methods with advantages in microscale analysis need to be resorted to.

The ReaxFF MD method differs from the traditional numerical simulation methods in that it considers the bond breaking and bond formation of molecules. It can obtain the microscopic changes and reaction products of materials, which has significant advantages in accurately simulating and analyzing surface reactions on the microscale [20]. ReaxFF MD has been widely used in recent years to study the damage mechanism of Kapton and other aerospace materials by AO [15,16,21,22,23,24]. First, the AO effect on Kapton, Teflon, and other polymer materials was studied using ReaxFF MD [21]. This study calculated the materials’ mass loss, temperature change, and erosion yield, which agreed well with the experimental data. The results indicated that the temperature rise at the material surface played a vital role in the AO erosion process. Then, the damage effects of AO energy and cumulative flux on aerospace metallic materials such as silver and aluminum were studied [15]. The results showed that the number of material defects and damage depth increased linearly with impact energy. Moreover, studies on the resistance to AO of nanocomposites such as PI/POSS with PI as a substrate have been carried out [16,22]. The findings showed that the PI exposure content on the surface of the PI/POSS system was a key parameter affecting the erosion yield of the material. The cage-like structure of POSS acted as a physical barrier and consumed most of the kinetic energy of AO, thus mitigating the erosion of the PI substrate by AO. The above studies show that the ReaxFF MD method is relatively mature in terms of techniques for studying the AO effect on Kapton films and has excellent potential for AO parameter-dependent studies. However, most previous studies only considered the vertical incidence and constant impact frequency of atomic oxygen. The AO incidence angle and flux density in space are diverse, which may cause different damage effects on aerospace materials. The ReaxFF MD method can provide a clearer understanding of the influence of AO parameter variation on the erosion mechanism and holds promise for resolving the inconsistency between airborne and ground test results.

In the ReaxFF MD simulation, the change in the incidence angle can be achieved by changing the velocity direction of atomic oxygen. In addition, the essence of the change in flux density is the change in AO impact frequency. To save computational resources, the effect of flux density can be studied qualitatively by increasing the AO dose rate to achieve an increase in the impact frequency in the ReaxFF MD simulation. The increase in dose rate needs to be kept at the same ratio as the increase in flux density in the experiment. Based on the above considerations, this study first conducted synergistic tests with a changing incidence angle and flux density using microwave plasma-based ground test equipment to obtain Kapton’s mass loss data. Then, univariate simulations of incidence angle and dose rate were performed using the ReaxFF MD simulation method. This revealed the effect patterns of the incidence angle and dose rate on the Kapton’s mass loss, temperature change, erosion yield, and generated products. The dependence of the AO effect on the incidence angle and flux density was further understood by the comparative analysis of simulation results and experimental data. The results show that the AO effect in the lower incidence angle range (0–60°) is independent of the incidence angle and depends only on the amount of impacted atomic oxygen; the AO effect in the higher incidence angle range (60–90°) is strongly dependent on both the incidence angle and the amount of impacted atomic oxygen. The AO effect at higher incidence angles has a surface stripping effect, which strips the rough raised macromolecules from the material surface and causes greater mass loss and a temperature rise. There is a critical value for the influence of flux density on the AO effect. Above this critical value, AO has a reduced erosive capacity due to a lower chance of participating in the reaction. The critical AO flux density obtained in this study is approximately 10 × 10^15^ atoms cm^−2^·s^−1^. The amount of each main product from the AO effect varies with the incidence angle and flux density. Nonetheless, the total content of the main products is essentially constant, around 70%. This work can improve the understanding of the parameter dependence of the AO effect and provide valuable information for the development of standards for ground simulation tests.

## 2. Methods

### 2.1. Experimental Method

#### 2.1.1. Ground-Based Test Device

A microwave plasma-type ground experimental device (AO-1, Feifan Plasma Technology Co., Ltd., Hefei, China) was used to conduct the AO exposure test. The principle of the test equipment has been reported in a previous study [24]. A schematic diagram of the ground-based atomic oxygen experimental setup is shown in Figure 1. This device used microwave discharge technology to ionize the incoming oxygen to form oxygen plasma. A magnetic field then accelerated the oxygen plasma to impact a negatively charged neutral target placed at a 45° tilt. The oxygen plasma was reduced to a neutral vertical downward atomic oxygen beam. The neutral target was composed of molybdenum metal with a diameter of 12 cm and was negatively biased by a 36 V DC-regulated power supply with a reflection efficiency of around 18%. The uniformity of the AO beam generated by this device was greater than 90%, and the energy was approximately 5 eV. The method of determining the neutral target’s reflection efficiency of the neutral target is described in the Appendix A. The flux density of AO was determined by both the target current (ion current) and the reflection efficiency of the target [25,26]. It was calculated according to the following equation:(1)$$\{\begin{array}{c} F_{i}=\frac{I_{i}}{{\mathrm{eA}}_{C}}\\ F_{\mathrm{AO}}=\eta F_{i} \end{array}$$
where $F_{\mathrm{AO}}$ is the atomic oxygen flux density, atoms cm^−2^·s^−1^; $F_{i}$ is the ion flux density, ions cm^−2^·s^−1^; η is the neutral target reflection efficiency; $I_{i}$ is the target current, A; e is the electron charge, 1.6 × 10^−19^ C; $A_{C}$ is the neutral target collecting pole area, cm^2^.

#### 2.1.2. Experimental Materials

The material used in this study was a Kapton film (produced by DuPont, Wilmington, DE, USA). The density of the material was 1.4 g/cm^3^, and the thickness was 50 μm. The film was cut into 20 × 20 mm square samples and ultrasonically cleaned in anhydrous ethanol for 10 min to remove any residual organic contaminants from the material’s surface. After cleaning, the films were dried naturally and prepared for use.

#### 2.1.3. Experimental Conditions and Procedures

This experiment aimed to investigate the effect of different AO parameters on the mass loss of Kapton films, including the incidence angle and flux density. According to Equation (1), the flux density was adjusted by proportionally increasing the target current (ion current), and the incidence angle was adjusted by changing the inclination angle of the Kapton film samples. A parametric study of the problem for the following cases was conducted:
Ten values of the incidence angle (θ = 0°, 10°, 20°, 30°, 40°, 50°, 60°, 70°, 80°, and 90°);Five values of the target current ($I_{i}\;$= 0.25 A, 0.5 A, 1.0 A, 1.5 A, 2.0 A).


The experiments were divided into five groups according to different target current values. Ten samples of Kapton films with different angles were placed on the AO exposure platform for each group of tests. In total, 50 samples were used. The AO flux densities for each group of experiments were calculated according to Equation (1) as 2.5, 5, 10, 15, 20 × 10^15^ atoms cm^−2^·s^−1^. The ratio of flux densities was 1, 2, 4, 6, 8. Each group of samples was exposed to AO for 10 h. The mass loss was calculated by weighing the sample before and after the test using an electronic analytical balance (x-56, accuracy 0.1 mg, Mettler Toledo, Zurich, Switzerland).

### 2.2. Simulation Method

#### 2.2.1. Reactive Molecular Dynamics

Reactive molecular dynamics (ReaxFF MD) was used to simulate the erosion process of Kapton by AO. Unlike traditional polymer force fields such as AMBER, CHARMM, CVFF, COMPASS, etc., ReaxFF is a bond-order-based force field that reasonably describes bond breakage and bond formation and is suitable for AO effect simulation [20,27]. In ReaxFF, the following expression is used to derive the forces on each atom:
(2)$$E_{\mathrm{system}}=E_{\mathrm{bond}}+E_{\mathrm{triple}}+E_{C2}+E_{\mathrm{val}}+E_{\mathrm{pen}}+E_{\mathrm{tors}}+E_{\mathrm{lp}}+E_{\mathrm{coa}}+E_{\mathrm{conj}}\;+E_{\mathrm{over}}+E_{\mathrm{under}}+E_{\mathrm{Hbond}}+E_{\mathrm{vdW}}+E_{\mathrm{coulomb}}$$
where $E_{\mathrm{system}}$ is the total system energy; $E_{\mathrm{bond}}$, $E_{\mathrm{triple}}$, $E_{C2}$ are the bond energy term, triple bond energy correction term, and triple bond energy penalty term, respectively; $E_{\mathrm{val}}$, $E_{\mathrm{pen}}$ are the bond angle energy term and bond angle energy penalty term, respectively; $E_{\mathrm{tors}}$, $E_{\mathrm{lp}}$, $E_{\mathrm{coa}}$, $E_{\mathrm{conj}}$ are the torsion angle energy term, lone pair electron energy term, three-body conjugate effect energy term, and four-body conjugate effect energy term, respectively; $E_{\mathrm{over}}$, $E_{\mathrm{under}}$ are over- and under-alignment energy correction terms; $E_{\mathrm{Hbond}}$, $E_{\mathrm{vdW}}$, $E_{\mathrm{coulomb}}$ are non-bonding hydrogen bonding interaction terms, van der Waals interaction terms, and Coulomb interaction terms, respectively.

Unlike the non-reactive force fields, the non-bonding interactions in ReaxFF are calculated between each pair of atoms without considering their connectivity. Shielding terms are used in the van der Waals and Coulomb energy terms to eliminate any excessive non-bonding interactions. A seven-degree taper function is used in these non-bonding interaction energies to eliminate any discontinuities [28]. In addition, the atomic charges can be determined using a geometrically relevant charge calculation scheme called the electronegativity equalization method (EEM) [29].

The bond order is then calculated using the interatomic distance to determine the interaction between atoms. The bond order expression is as follows:(3)$${\mathrm{BO}}_{\mathrm{ij}}^{i}={\mathrm{BO}}_{\mathrm{ij}}^{\sigma }+{\mathrm{BO}}_{\mathrm{ij}}^{\pi }+{\mathrm{BO}}_{\mathrm{ij}}^{\pi \pi }$$
where ${\mathrm{BO}}_{\mathrm{ij}}^{i}$ is the bond order between atom i and atom j. ${\mathrm{BO}}_{\mathrm{ij}}^{\sigma }$, ${\mathrm{BO}}_{\mathrm{ij}}^{\pi }$, and ${\mathrm{BO}}_{\mathrm{ij}}^{\pi \pi }$ are the bond order contributions for a sigma bond, pi bond, and double-pi bond, respectively.

In this study, the force field parameters of carbon, hydrogen, oxygen, and nitrogen from Rahnamoun et al. were used to perform AO impact simulations of Kapton films [21]. The force field parameters have been successfully applied to molecular dynamics simulations of the AO effect on various polymers [16,30].

#### 2.2.2. Simulation Details

Since the ReaxFF MD simulation method is computationally expensive, and the focus of this study is not on the size effect of nanoparticles, the computational approach in this study drew on the work of Rahnamoun and Rahmani et al. [21,22]. The steps to build the Kapton structure are shown in Figure 2. First, a polyimide molecule (C_22_H_12_N_2_O_5_) was created, where H is represented by white atoms, N by blue atoms, C by gray atoms, and O by red atoms. Then, an amorphous model containing 140 PI monomers with a density of 0.1 g/cm^3^ was constructed. After a 1 ns NPT (constant atomic number, N; constant pressure, P; constant temperature, T) simulation at room temperature (298 K) and atmospheric pressure, the material density was stabilized at 1.37 g/cm^3^, close to the test material density. Finally, the periodicity of the model in the z-direction was canceled, and the vacuum region above the material surface was constructed by increasing the box z-direction size to provide space for subsequent atomic oxygen insertion. The Nosé–Hoover thermostat and Berendsen barostat were used for temperature and pressure control, respectively [31]. The dimensions of the material region of the final model were 40.3 Å × 40.3 Å × 50.3 Å, with a total atomic number of 5740.

After the model was built, AO impact simulations were performed. First, the model’s x- and y-direction boundaries were set to periodic boundaries, and the z-direction boundaries were set to fixed boundaries. The z-directional size of the box was increased to facilitate statistical products. Then, the energy minimization of the system was performed using the conjugate gradient method [32], followed by NVT (constant atomic number, N; constant volume, V; constant temperature, T) simulations at 300 K using the Langevin method for 30 ps until the system temperature stabilized. Finally, the thermostat was removed, and NVE (constant atomic number, N; constant volume, V; constant energy, E) simulations were run to fix the volume and energy of the system while allowing for an increase in temperature. Oxygen atoms were randomly inserted in the region 60 Å above the material surface with a velocity size of 7.8 km/s (energy of 5 eV, consistent with the experiment). All the above operations were performed in the LAMMPS package [33,34].

The incidence angle and flux density are essentially the impact direction and frequency of atomic oxygen. In this work, the influences of the incidence angle and flux density on the AO effect were investigated by changing the velocity direction and dose rate of AO, respectively, using the ReaxFF MD method. First, the mass loss, temperature change, erosion yield, and products of Kapton films irradiated for 60 ps (cumulative flux 2E15 atoms/cm^2^) by AO, at a constant dose rate (5 atoms/ps) and changing velocity direction (0–85° at 5° intervals), were calculated to analyze the incidence angle dependence of the AO effect. The results for Kapton films with a constant AO velocity direction (0°) and changing dose rate (5, 10, 20, 30, 40 atoms/ps) were also calculated to analyze the flux density dependence of the AO effect. In the case of varying incidence angles, the x- and z-direction velocity magnitudes of atomic oxygen were modified to change the incidence angle; in the case of varying dose rates, the insertion frequency of atomic oxygen was adjusted to increase the dose rate proportionally. The critical information for 23 calculation cases is presented in Appendix A.

The reactive force field parameters used in the above AO erosion simulations were taken from the work of Rahnamoun et al. [21]. The time step and the cut-off distance were set to 0.1 fs and 9 Å. To prevent the whole model from “drifting”, the lower surface of the material (thickness of around 5 Å) was fixed. In addition, the “fix wall/reflect” command in LAMMPS was used to give the effect of a reflective wall on the bottom surface of the box, preventing the atoms from crossing the lower boundary. This method allowed for more accurate statistics of the mass change of the material. During the simulation, the mass and temperature changes of the material were output in real time, and the products of detachment were counted using “fix reax/c/species”. The trajectory files were saved every 100 fs, and the atomic motion during the AO erosion simulation was visualized using the OVITO software [35]. This was an essential aid for the mechanistic analysis of the AO erosion process.

## 3. Results and Discussion

### 3.1. Ground-Based Experiment

This section evaluates the mass loss of Kapton films under a bivariate AO incidence angle and flux density. Figure 3 shows the mass loss data of Kapton films after 10 h of AO irradiation at different flux densities and incidence angles. The mass loss at all incidence angles showed an increasing trend with increasing flux density, which is consistent with the results of Kim et al. regarding the variation in dose rate parameters [16]. However, the mass loss curves for the incident angles of 0–60° and 70–90° showed different patterns. The mass loss curves for the incident angles of 0–60° increased approximately linearly with increasing flux density and then leveled off. The mass loss curves for the incident angles of 70–90° showed a linear increase with increasing flux density.

To better understand the flux density effect on the AO effect, flux density and mass loss data were normalized. Figure 4 shows Kapton films’ mass loss versus flux density at different incidence angles. The results showed that in the lower flux density (2.5 − 10 × 10^15^ atoms cm^−2^·s^−1^) range, the mass loss of Kapton films was proportional to the flux density. In contrast, at higher flux densities (15−20 × 10^15^ atoms cm^−2^·s^−1^), the mass loss values deviated below the positive scale line, and the deviation became more significant as the flux density increased. This result suggests that the mechanism of damage to Kapton films by AO at lower flux densities may be the same. The mass loss was only related to the amount of atomic oxygen impinging on the material surface. In comparison, at higher flux densities, the mass loss of Kapton films was also influenced by other mechanisms of AO action. This unknown mechanism of action was thoroughly investigated in a subsequent ReaxFF MD simulation.

To analyze the influence of the incidence angle on the AO effect more intuitively, the mass loss was normalized using the 0° incidence angle results as a benchmark. Figure 5 shows Kapton films’ normalized mass loss versus incidence angle at different flux densities. The results showed that the mass loss at different flux densities displayed the same pattern with the change in the incidence angle. In the lower incidence angle range (0–60°), the mass loss due to atomic oxygen conformed to the cosθ curve, which is the same as the results of Tagawa et al. [19]. In comparison, the values of mass loss at higher incidence angles (70–90°) were higher than the cosθ curve. The results suggest that the mechanism of the erosion of Kapton films by atomic oxygen may differ for different incidence angle ranges. The mass loss due to atomic oxygen at 0–60° may be related only to the accumulated flux (amount of atomic oxygen). In contrast, there were other influencing factors for the mass loss due to atomic oxygen at 70–90°. In the simulation phase, a more detailed study of the parametric influence of the incidence angle was developed using ReaxFF MD.

### 3.2. ReaxFF MD Simulation

#### 3.2.1. Effect of the Incidence Angle

Figure 6 shows snapshots of ReaxFF MD simulations of the atomic oxygen erosion of Kapton material for different incidence angles. The atomic oxygen irradiation of 60 ps (300 oxygen atoms) was performed for each calculation case, and the material’s mass loss, detachment products, erosion yield, and temperature variation were determined. The incidence angle dependence of the AO effect was investigated by analyzing the variability of the erosion of the material for different incidence angles of atomic oxygen.

##### Mass Loss

This subsection quantifies the simulation results of the mass loss of Kapton films for the case of varying AO incidence angles. Figure 7 shows the mass change of Kapton material under the impacted atomic oxygen at different incidence angles. The mass of the material at all incidence angles showed a trend of increasing and then decreasing, which is the same as in the results of the atomic oxygen tests and simulations of Kapton materials performed by other researchers [21,36]. The initial physical adsorption of atomic oxygen impacting the material surface led to an increase in mass. As the amount of impacted atomic oxygen increased, the temperature of the material rose to initiate pyrolysis and it reacted with atomic oxygen to release the products, which led to a loss of mass in the material. Figure 8 shows the Kapton mass loss data caused by the atomic oxygen impact at different time points. After 30 ps, the mass loss showed a linearly increasing trend for each incidence angle, so data from 30 to 60 ps (cumulative flux of 1−2 × 10^15^ atoms/cm^2^) were selected to analyze the effect of the incidence angle.

Figure 9 shows Kapton’s normalized mass loss at different AO incidence angles for each time node within 30–60 ps (cumulative flux 1−2 × 10^15^ atoms/cm^2^). The process can be divided into three stages based on the amount of atomic oxygen: in the first stage (30–36 ps), the number of impacted atoms was small, and the mass loss data for each incidence angle were distributed on both sides of the cosθ, but the distribution was scattered; in the second stage (42–48 ps), the number of impacted atoms increased and the mass loss results for each incidence angle were more compactly distributed near the cosθ curve; in the third stage (54–60 ps), enough oxygen atoms hit the material surface and fully interacted with the material. At this time, the data distribution of mass loss at each incident angle showed more apparent characteristics, i.e., the mass loss curve conformed to the cosθ curve when the incident angle was less than 60°, and the mass loss data for the incident angle larger than 60° were all located above the cosθ curve.

The simulated mass loss results of the Kapton when changing the incidence angle of atomic oxygen showed essentially the same pattern as the experimental results. Mass loss values caused by atomic oxygen at an incidence angle greater than 60° deviated from the cosθ curve. The deviation of the mass loss data from the cosθ curve was more significant for an experimental incidence angle above 60°, which can be attributed to the fact that the amount of atomic oxygen in the experiment was much higher than that in the simulation, resulting in a more pronounced feature.

To investigate the reasons for the above mass loss characteristics due to the change in incidence angle, the mass loss rate information at each incidence angle was first calculated based on the data from 30 to 60 ps, as shown in Figure 10. Then, the data on the amount of impacted atomic oxygen and its mass loss rate were normalized to 0° and used to investigate the dependence of the mass loss on the amount of atomic oxygen for different incidence angles, as shown in Figure 11.

Figure 11 presents the same characteristics as Figure 10. The value of the mass loss rate of the Kapton material due to atomic oxygen at incidence angles greater than 60° was higher than the cosθ curve. From the results in Figure 11, it can be seen that the mass loss rate of Kapton due to atomic oxygen at incidence angles less than 60° was proportional to the amount of impacted atomic oxygen, and the mass loss rate caused by atomic oxygen at incidence angles greater than 60° was higher than the positive scale line. This suggests that the erosion of Kapton by atomic oxygen at incidence angles less than 60° depended only on the cumulative flux of atomic oxygen, independent of the incidence angle, which confirms the speculation of Tagawa et al. [19]. In contrast, when the incidence angle of atomic oxygen was greater than 60°, this study found that the erosion of atomic oxygen was not only dependent on the accumulated flux of atomic oxygen, but there were also unique mechanisms leading to more severe erosion effects.

To understand more clearly the differences in the erosion mechanisms of atomic oxygen at different incidence angles, the mass loss and mass loss rate caused by atomic oxygen were calculated using the mass loss and mass loss rate divided by the effective amount of impacted atomic oxygen, which are shown in Figure 12 and Figure 13, respectively. The mass loss caused by single atomic oxygen was the same for an incidence angle less than 60° and increased significantly for an incidence angle greater than 60°. The mass loss rate contributed by individual atomic oxygen also showed the same trend and was more distinctive.

Simulation and experimental results showed that Kapton films’ atomic oxygen erosion mechanisms differed in incidence angle ranges. The erosion effect of atomic oxygen for incidence angles less than 60° was independent of the incidence angle and was only related to the cumulative flux of atomic oxygen. The superposition of the number of single atomic oxygen effects led to an increase in the mass loss of the Kapton material. The erosion effect of atomic oxygen on Kapton material for incidence angles greater than 60° depended on the incidence angle and became more severe with the increasing incidence angle.

##### Products and Erosion Yield

Another indicator that characterizes the ability of atomic oxygen to erode is the erosion yield. The erosion yield was defined as the average mass loss of the material divided by the number of oxygen atoms impinging on the material [37]. To ensure consistency with the test method’s units of erosion yield, the erosion yield was calculated in the ReaxFF MD method by using the total mass of the products detached from materials divided by the amount of impacted atomic oxygen [21,38].

In this section, the types and amounts of products are presented. The products broke away from the material’s surface after being attacked by atomic oxygen. The erosion yield of atomic oxygen was obtained by dividing the total mass of products by the number of impacted oxygen atoms at each angle of incidence. Figure 14 shows the information on erosion products at different incidence angles, including the number and content of products and the total content of the five main products (CO, O_2_, H_2_O, OH, NO). It can be seen that the number of products did not change significantly when the incidence angle was less than 60°, and the number decreased significantly after it was greater than 60°. The content of various products changed with the variety of incidence angles, but the total content of the main products was unchanged, and the average value was 72.35%. Figure 15 shows the calculated erosion yield of atomic oxygen for each incidence angle. The results were consistent with the mass loss pattern in the previous subsection. The erosion ability of atomic oxygen at incidence angles less than 60° was essentially the same; the erosion ability of atomic oxygen at incidence angles greater than 60° increased with the increasing incidence angle.

Combining the above results of mass loss and erosion yield caused by single atomic oxygen, the incidence angle dependence of the AO effect was demonstrated. The AO erosion mechanism was the same for incidence angles less than 60°. Atomic oxygen at incidence angles greater than 60° exhibited a more considerable erosion capability, which increased with the increasing incidence angle. This feature became more pronounced as the amount of impacted atomic oxygen increased. This explains why the mass loss results caused by atomic oxygen with high flux and a high incidence angle in the experiment were more skewed above the cosθ curve compared to the mass loss results in the simulation. Further, the reasons for this incidence angle dependence are explored below.

##### Temperature Variation

Ground-based experiments and simulation studies have shown that the temperature variation of the material after atomic oxygen impact is a critical factor in the generation of the AO effect [7,21,39,40]. Analysis of the variability of material temperature due to incidence angle variation can help to investigate the reason for the incidence angle dependence of the AO effect.

Figure 16 illustrates the Kapton’s temperature variation for the AO irradiation process at different incidence angles. As the simulation time increased, the temperature of the Kapton material under atomic oxygen impact at different incidence angles showed an increasing trend. In the same way as in the mass loss data, the temperature rise rate of the Kapton material at each incidence angle within 30–60 ps was calculated. As shown in Figure 17, the temperature rise rate of the material gradually decreased as the incidence angle increased and gradually deviated above the cosθ curve. Figure 18 plots the normalized amount of atomic oxygen versus the standard temperature rise rate for different incidence angles. At lower incidence angles, the temperature rise rate of the material was proportional to the number of oxygen atoms. As the incidence angle increased, the temperature rise rate was gradually higher than the positive scale line. This indicates that the temperature rise caused by the AO effect on the material at higher incidence angles was equally dependent on the incidence angle.

In order to obtain a clearer understanding of the temperature rise variability caused by the atomic oxygen at different incidence angles, the normalized temperature rise and temperature rise rate were calculated using the temperature rise and temperature rise rate divided by the effective amount of impacted atomic oxygen, which are shown in Figure 19 and Figure 20 respectively. The temperature rise and the rate of temperature rise caused by single atomic oxygen were flat in the lower incidence range. In contrast, the temperature rise and the rate of temperature rise caused by atomic oxygen at higher incidence angles showed an increasing trend with increasing incidence angles. In particular, the temperature increase was more apparent when the incidence angle was greater than 60°.

The temperature rise data summarized above showed the same trend as the previously discussed mass loss data with the incidence angle, which is consistent with the conclusions of previous studies that the erosion mechanism of atomic oxygen is inextricably linked to the temperature variation of the material [7,21,39,40]. In the following, the reasons for the dependence on the incidence angle of atomic oxygen are investigated in terms of the temperature statistics of the material. It is well known that temperature is a measure of the intensity of the thermal motion of atoms. The more intense the atomic thermal action is, the higher the temperature is. The molecular dynamics approach likewise establishes the relationship between material temperature and atomic motion velocity for three-dimensional systems based on statistical mechanics [34,41], as shown in the following equation:(4)$$\frac{3}{2}{\mathrm{Nk}}_{B}T=\overset{N}{\underset{i=1}{\sum }}\frac{1}{2}m_{i}v_{i}{}^{2}$$
where the right-hand side of the formula is the total kinetic energy of the atomic group. N is the total number of atoms, $m_{i}$ is the atomic mass, and $v_{i}$ is the atomic velocity. On the left side of the formula, $k_{B}$ is the Boltzmann constant, and T is the average temperature of the atomic group. From the equation, it is clear that the increase in temperature is due to the increased velocity of the atoms in the statistical region of the material. It is reasonable to speculate that atomic oxygen with higher incident angles caused more significant temperature rises because the atoms on the material surface moved more violently.

Figure 21 and Figure 22 show the evolution of the initial phase of the atomic oxygen impact on the Kapton material surface for 0° and 85° incidence angles. It can be seen that the 0° atomic oxygen impacting the material surface first underwent physisorption and was deposited on the material surface, at which point no reaction occurred. As the number of atomic oxygens increased, the material gradually underwent thermal decomposition due to the temperature rise. At the same time, the reaction with oxygen atoms generated CO, OH, O_2_, H_2_O, and other products. These products broke away from the material, causing mass loss. Unlike the erosion mechanism of atomic oxygen at 0°, when the atomic oxygen hit the material surface approximately horizontally, it directly stripped the raised macromolecules of the surface material. The atoms on the surface underwent violent motion and subsequently detached from the surface, causing the mass loss of the material, accompanied by an increase in the temperature of the material.

This experiment found the same mass loss law as the ground test results using the ReaxFF MD simulation. Then, the incidence angle dependence of the AO effect was clarified by calculating the erosion yield and the temperature rise caused in the Kapton material for different incidence angles in the simulation results. Finally, the possible causes of the incidence angle dependence of the AO effect were found from the statistical principle based on the characteristics of the material temperature. To summarize the above analysis, it is concluded that the AO effect in the lower incidence angle range (0–60°) was independent of the incidence angle and depended only on the amount of impacted atomic oxygen; the atomic oxygen effect in the higher incidence angle range (60–90°) was strongly dependent on both the incidence angle and the amount of impacted atomic oxygen. There was a surface stripping effect of the atomic oxygen at higher incidence angles when it hit the Kapton material. The atomic oxygen stripped the rough, raised macromolecules from the material surface, causing greater mass loss and a temperature rise. This surface stripping effect became more significant as the incidence angle and the amount of impacted atomic oxygen increased.

#### 3.2.2. Effect of the Dose Rate

Figure 23 shows snapshots of the ReaxFF MD simulations of the Kapton material eroded by atomic oxygen at different dose rates. Each calculation was performed with 30 ps of atomic oxygen irradiation, and the material’s mass loss, detachment products, erosion yield, and temperature variation were calculated. The dose rate dependence of the AO effect was investigated by analyzing the variability of the erosion of the material with different dose rates of atomic oxygen.

##### Mass Loss

This subsection quantifies the simulation results of Kapton’s mass loss for the case of changing the AO dose rate. Figure 24 illustrates the mass change of the Kapton under different dose rates of atomic oxygen impact. At the initial stage of atomic oxygen impact, the mass of the material appeared to increase for different dose rate conditions. As the simulation time grew, the mass changes at different dose rates showed a decreasing trend. This was the same trend as the simulated one for the case of the changing incidence angle. In particular, the material masses of the different dose rate conditions in the initial stage grew differently. The larger the dose rate, the more significantly the mass increased, which may be related to the change in the mechanism of the AO effect at higher dose rates. Figure 25 shows the mass loss of the Kapton material after irradiation with atomic oxygen at different dose rates for 30 ps, indicating that the trend of increasing mass loss slowed down with the increasing dose rate, which showed the same characteristics as the mass loss curve in the experiment. Subsequently, the dose rate and the mass loss were normalized to investigate the relationship between the increasing dose rate and the mass loss, as shown in Figure 26. It can be seen that the mass loss was proportional to the dose rate at lower AO impact density conditions. However, the mass loss due to atomic oxygen impact at dose rates above 10 atoms/ps was lower than the positive proportional line.

To investigate the effect of a high impact frequency (high dose rate) on the AO effect, the trajectories of the initial phase (5 ps) of the atomic oxygen impact on the material at different dose rates were compared and analyzed. As shown in Figure 27, the amount of atomic oxygen deposited on the material surface gradually increased with the increasing dose rate, which may have hindered the subsequent contact reaction of atomic oxygen with the material and led to a decrease in the amount of atomic oxygen reacted. In contrast, the atomic oxygen at lower dose rates was less dense. It could fully interact with the material so that the resulting mass loss would be higher. Subsequent calculations of the erosion yield of single atomic oxygen verified these speculations. Figure 28 shows the proportion of atomic oxygen involved in the reaction. It can be seen that the quantity percentage of atomic oxygen involved in the reaction decreased at dose rates higher than 10 atoms/ps. The reason for the non-participation of atomic oxygen in the reaction may be the reflection of atomic oxygen. The deposition of surface oxygen atoms increased the surface density, increasing the chance of reflection. In addition, there were more surface products after the atomic oxygen hit the material at high dose rates. Some oxygen atoms hit the products and did not react effectively with the material.

##### Products and Erosion Yield

The same data processing steps were performed for the dose rate calculations. The erosion yield was calculated after the erosion products were counted. Figure 29 shows the product information after 30 ps of atomic oxygen impact at different dose rates. It can be seen that the number of products increased as the dose rate increased, with a significant increase in CO, a slight increase in other organic small molecules and intermediate products, a gradual decrease in OH, H_2_O, and O_2_, and very little change in NO. However, the total content of these main products remained the same, with an average value of 68.12%.

Figure 30 illustrates the erosion yield of the Kapton under atomic oxygen impact at different dose rates. The erosion yield did not change significantly for lower dose rates (5–10 atoms/ps) and decreased significantly for higher dose rates (20–40 atoms/ps), consistent with the previous discussion. On the one hand, the high density of atomic oxygen accumulated on the material surface hindered the subsequent contact reaction of atomic oxygen. On the other hand, the increase in the likelihood of atomic oxygen reflection led to a decrease in the amount of atomic oxygen involved in the reaction.

##### Temperature Variation

Figure 31 illustrates the Kapton’s temperature variation for the AO irradiation process at different dose rates. It can be seen that the temperature increased at different dose rates of atomic oxygen and showed a linear trend. The temperature rise rates at different dose rates were normalized, as shown in Figure 32. It can be seen that at lower dose rates, the temperature rise rate was proportional to the dose rate. At higher dose rates, the temperature rise rate was biased below the positive scale line, which was the same characteristic observed in the mass loss curves in the experiments and simulations. It can be seen that the impact effect of different dose rates of atomic oxygen was different, and the temperature rise caused by the impact of atomic oxygen at high dose rates was lower than that of atomic oxygen at low dose rates. Once again, it is verified that the effect of atomic oxygen impact at high dose rates was not simply a quantitative superposition of the single atomic oxygen effect, which led to a decrease in erosion capacity due to a change in the mechanism of action.

Since the essence of both a variable dose rate and variable flux density is to change the impact frequency of atomic oxygen, and the above simulation results obtained with a variable dose rate of atomic oxygen had the same pattern as the experimental results obtained with variable flux density, the simulation results could be used to interpret the experimental results. The following conclusions were obtained: the flux density of atomic oxygen had a specific effect on the damage of the Kapton material. When the flux density exceeded a certain limit, the damaging effect of atomic oxygen was not simply a quantitative superposition of the single AO effect. The decrease in the number of atomic oxygens involved in the reaction at high flux densities may have led to a lower erosion yield, so the damaging effect on the material of atomic oxygen at high flux densities was lower than that at low flux densities.

## 4. Conclusions

In this paper, the dependence of the incidence angle and flux density in the damaging effect of atomic oxygen on Kapton film was investigated by ground-based experiments in combination with ReaxFF MD simulations, with the following conclusions:
The mechanism of atomic oxygen erosion on Kapton films differs for different incidence angle ranges. The AO effect in the lower incidence angle range (0–60°) is independent of the incidence angle and depends only on the amount of impacted atomic oxygen; the AO effect in the higher incidence angle range (60–90°) is strongly dependent on both the incidence angle and the amount of impacted atomic oxygen. The AO effect at higher incidence angles has a surface stripping effect, which strips the rough, raised macromolecules from the material surface and causes greater mass loss and a temperature rise. This stripping effect becomes more significant with the increasing incidence angle and amount of impacted atomic oxygen. The findings on the incidence angle can help to improve the surface structure of spacecraft and develop new structural materials resistant to atomic oxygen erosion. It is possible to design a material with an arc-shaped surface structure, so that the incidence angle of atomic oxygen is only within the lower incidence angle range (0–60°). This may avoid the surface peeling effect and reduce the AO effect.There is a critical value for the effect of flux density on the AO effect. Below the critical flux density, the AO effect is a simple quantitative superposition of the single AO damage effect; above the critical flux density, the damaging effect caused by atomic oxygen is diminished, probably due to the reduced probability that high densities of atomic oxygen are involved in the reaction. The critical AO flux density obtained in this study is approximately 10 × 10^15^ atoms cm^−2^·s^−1^. The above conclusions suggest that shortening the experimental time by increasing the flux density exponentially may be effective only in a certain flux density range, which provides some reference significance for the development of accelerated test standards for ground simulation tests.Changes in the angle of incidence and dose rate have little effect on the total content of the main products of the AO effect, averaging around 70%. The use of ReaxFF MD for the statistics of product species and their contents in the AO effect provides an opportunity to estimate the erosion yield of atomic oxygen for new materials. This can complement traditional material characterization methods, contributing to a better understanding of the damage process of the AO effect.

## Figures and Tables

**Figure 1 polymers-14-05444-f001:**
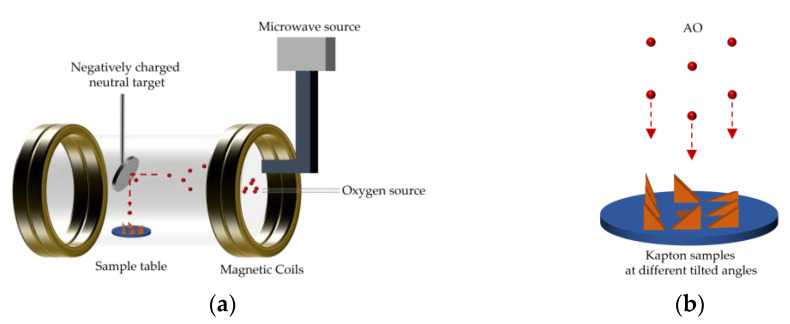
Schematic diagram of ground-based atomic oxygen experimental setup: (**a**) test device; (**b**) atomic oxygen erosion.

**Figure 2 polymers-14-05444-f002:**
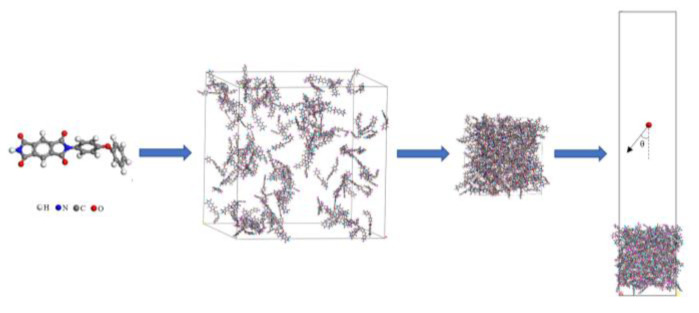
Modeling steps of Kapton structure.

**Figure 3 polymers-14-05444-f003:**
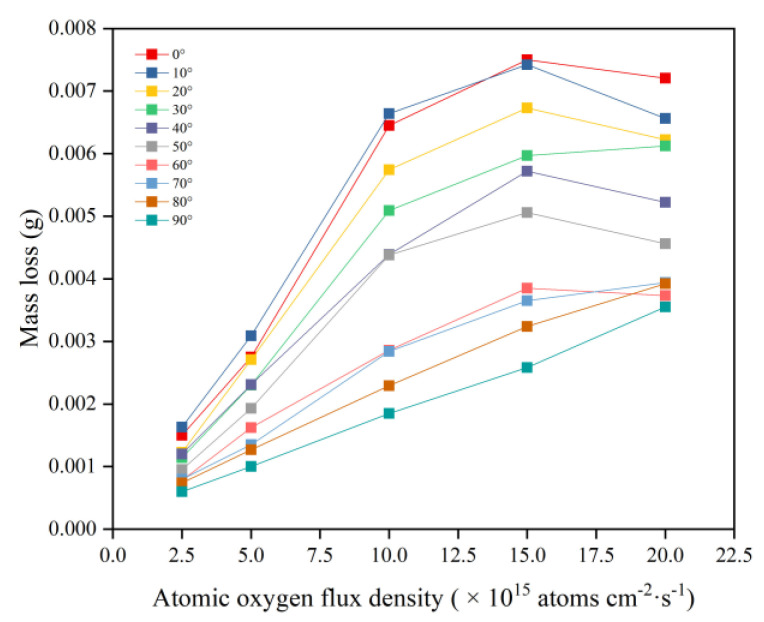
Mass loss of Kapton films at different flux densities and incidence angles.

**Figure 4 polymers-14-05444-f004:**
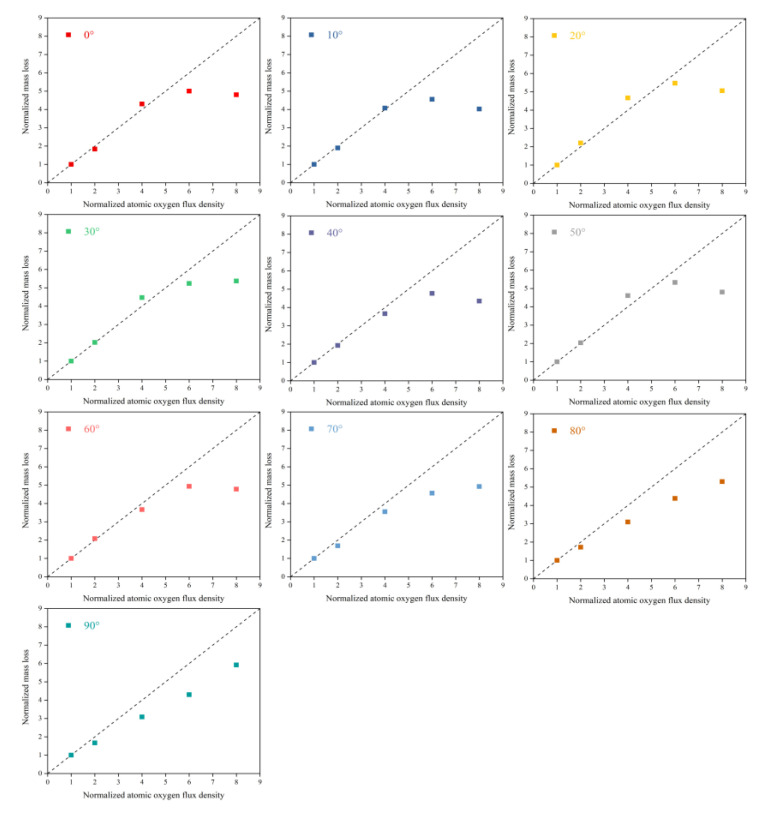
Normalized mass loss of Kapton films versus flux density at different incidence angles.

**Figure 5 polymers-14-05444-f005:**
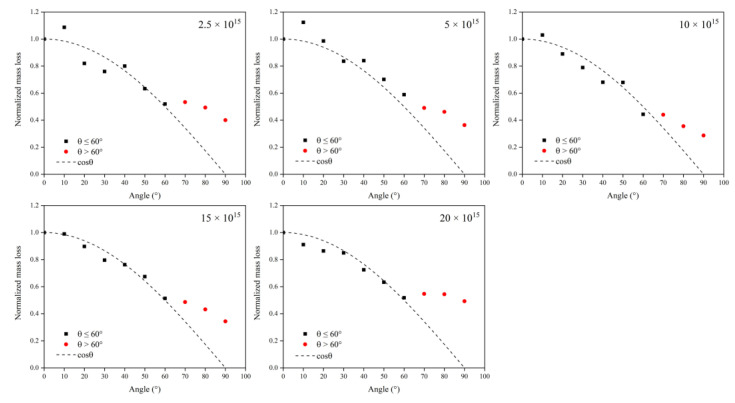
Normalized mass loss of Kapton films versus incidence angle at different flux densities.

**Figure 6 polymers-14-05444-f006:**
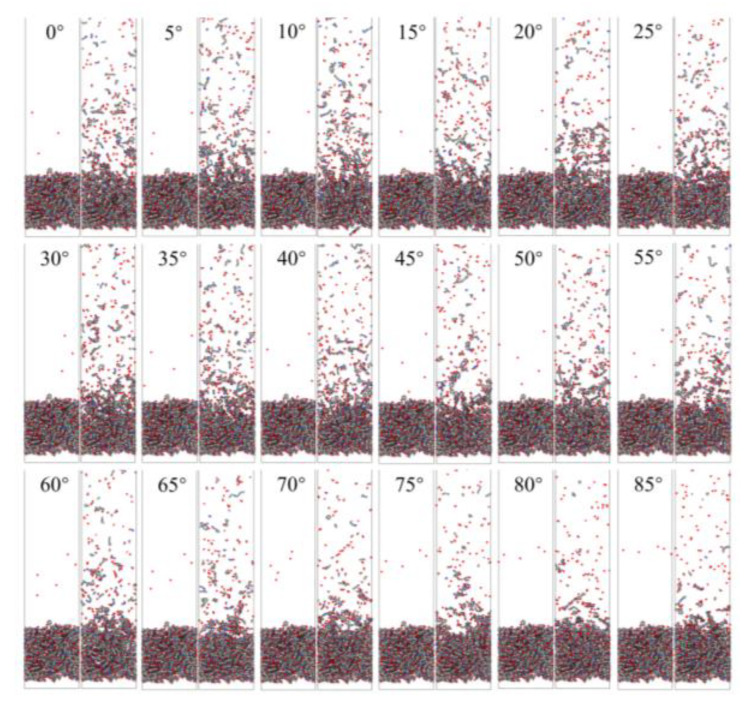
Snapshots of AO impacting Kapton at different incidence angles during the initial phase (1 ps) and the termination phase (60 ps).

**Figure 7 polymers-14-05444-f007:**
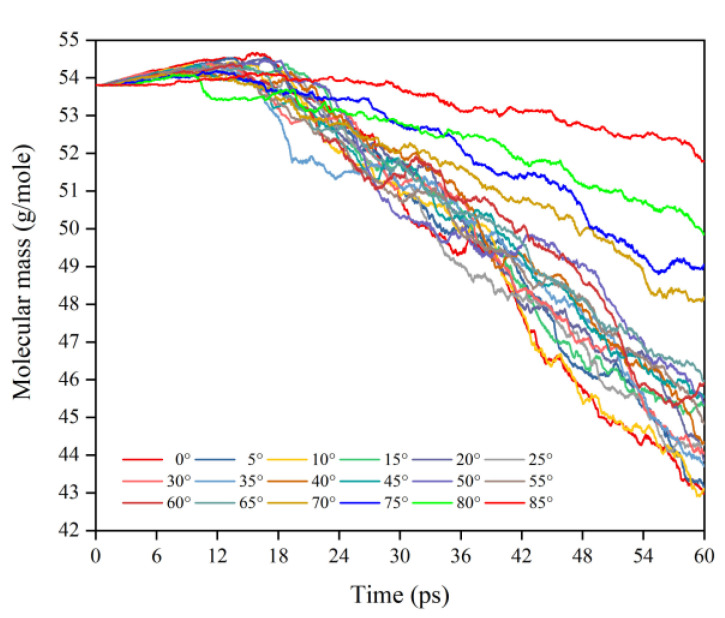
Mass changes of the Kapton during atomic oxygen impact at different incidence angles.

**Figure 8 polymers-14-05444-f008:**
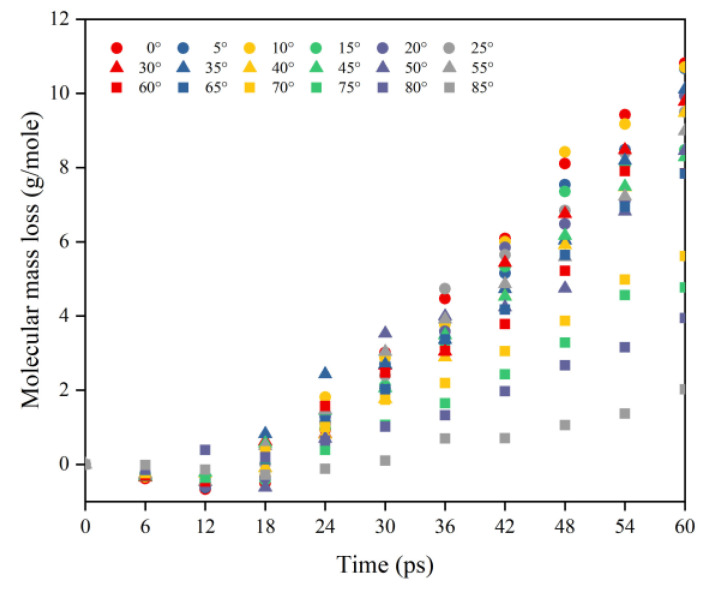
Mass loss of the Kapton during atomic oxygen impact at different incidence angles.

**Figure 9 polymers-14-05444-f009:**
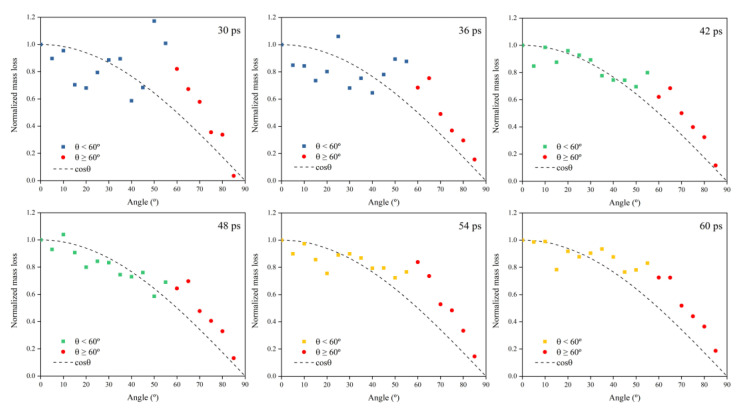
Normalized mass loss of the Kapton during atomic oxygen impact at different incidence angles.

**Figure 10 polymers-14-05444-f010:**
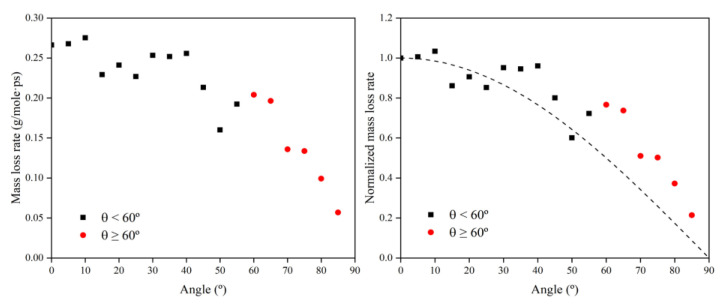
Mass loss rate information of the Kapton during atomic oxygen impact at different incidence angles.

**Figure 11 polymers-14-05444-f011:**
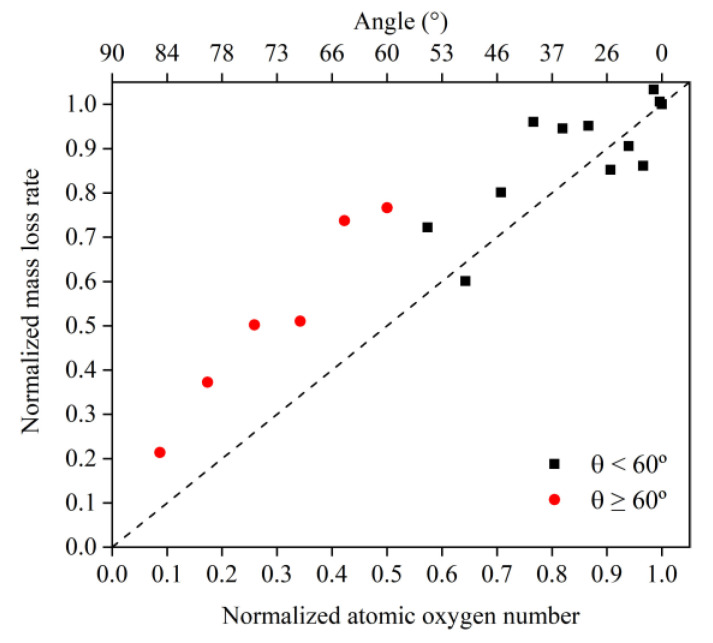
Normalized mass loss rate of the Kapton under different amounts (incidence angles) of impacted atomic oxygen.

**Figure 12 polymers-14-05444-f012:**
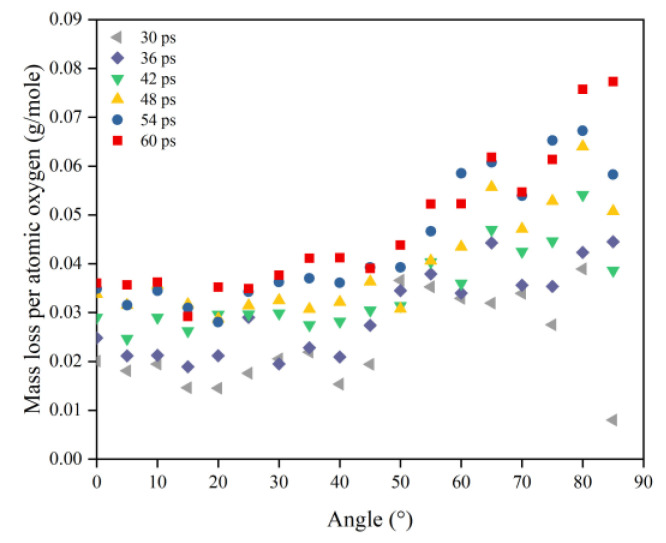
Mass loss of the Kapton caused by atomic oxygen at different incident angles at different time nodes.

**Figure 13 polymers-14-05444-f013:**
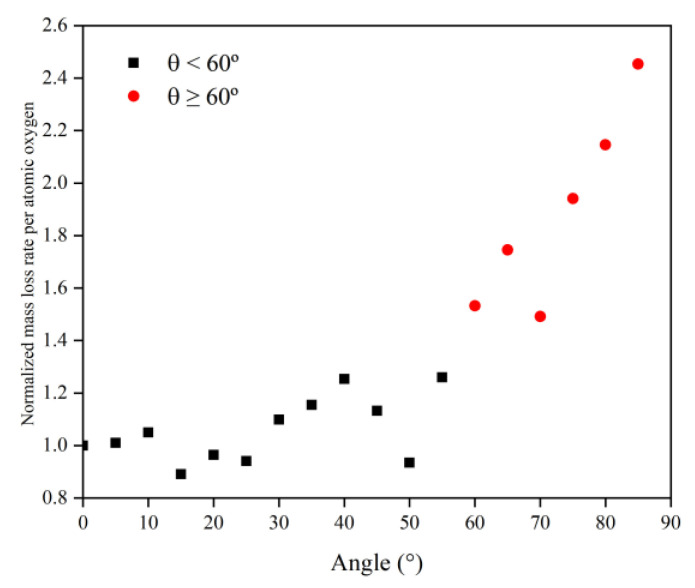
Mass loss rate of the Kapton caused by atomic oxygen at different incident angles.

**Figure 14 polymers-14-05444-f014:**
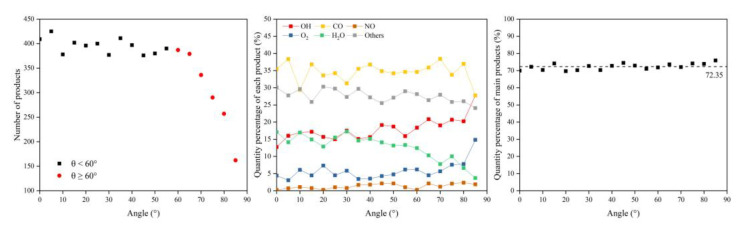
Product information of the Kapton after atomic oxygen impact at different incidence angles.

**Figure 15 polymers-14-05444-f015:**
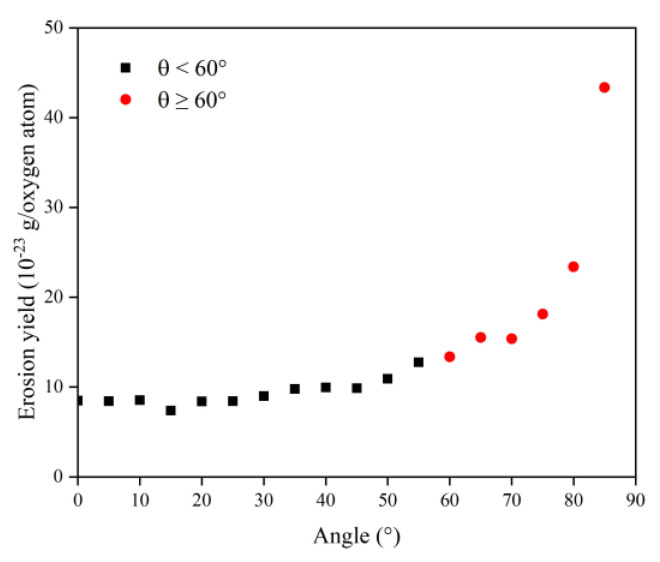
Erosion yield of the Kapton under atomic oxygen impact at different incidence angles.

**Figure 16 polymers-14-05444-f016:**
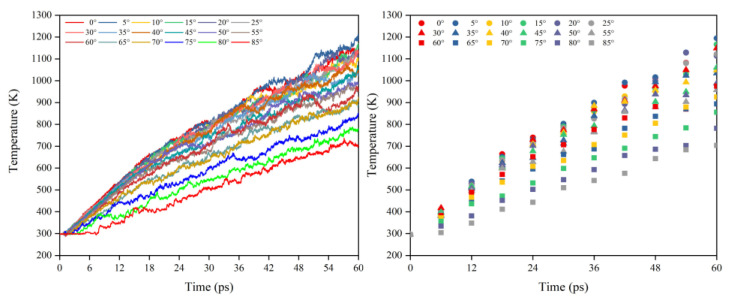
Temperature variation of the Kapton under atomic oxygen impact with different incidence angles.

**Figure 17 polymers-14-05444-f017:**
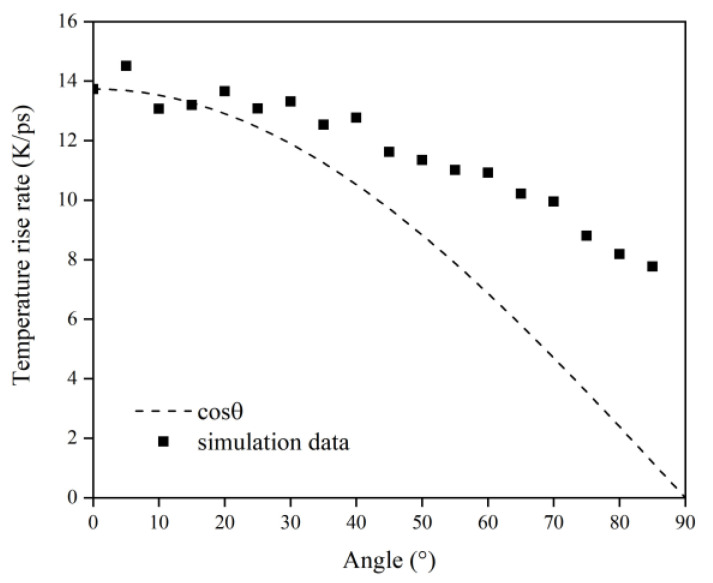
Temperature rise rate of the Kapton under atomic oxygen impact with different incidence angles.

**Figure 18 polymers-14-05444-f018:**
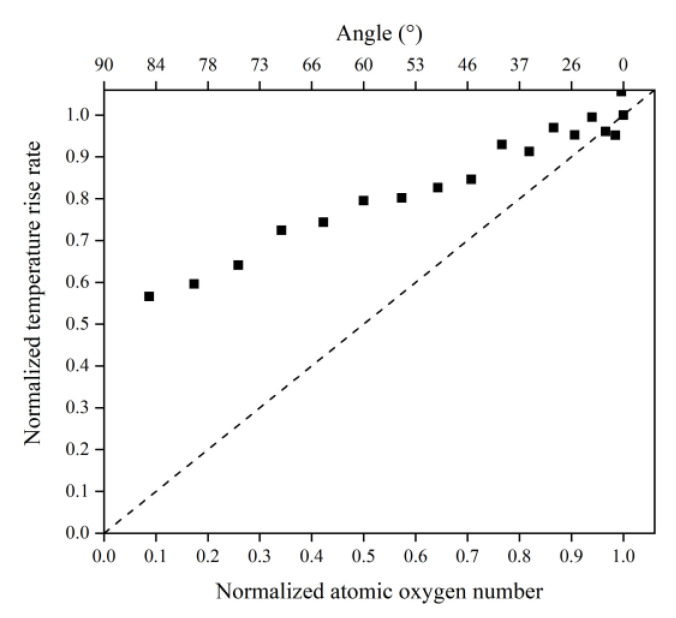
Normalized temperature rise rate of the Kapton under different amounts (incidence angles) of atomic oxygen impact.

**Figure 19 polymers-14-05444-f019:**
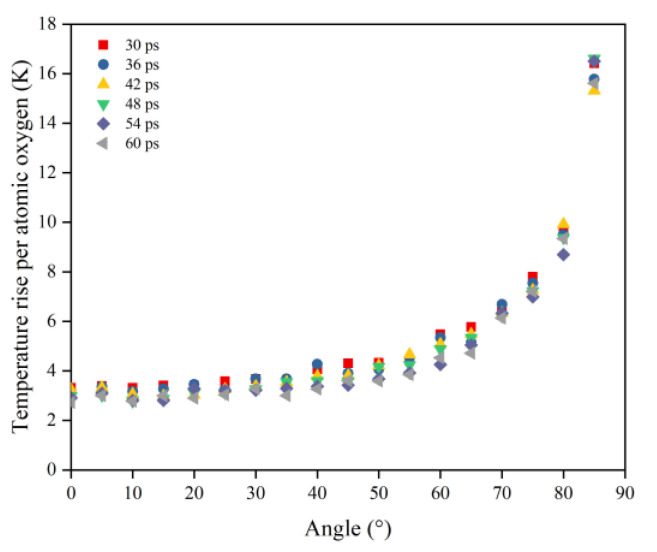
Temperature rise of the Kapton caused by per atomic oxygen at different incident angles at different time nodes.

**Figure 20 polymers-14-05444-f020:**
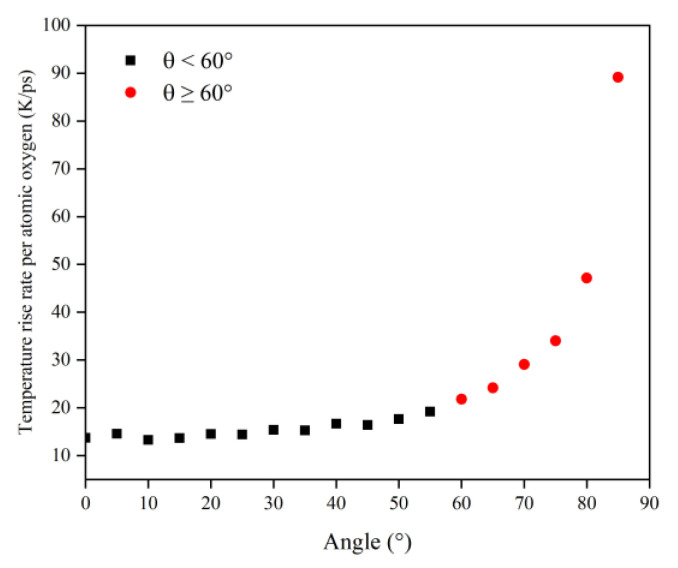
Temperature rise rate of the Kapton caused by atomic oxygen at different incident angles.

**Figure 21 polymers-14-05444-f021:**
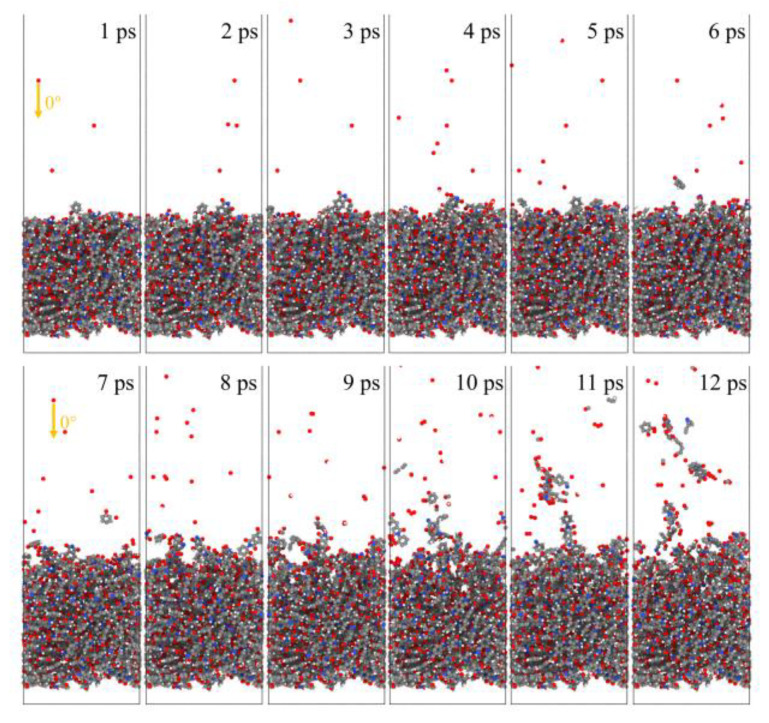
Evolution of the initial phase of atomic oxygen impacting Kapton material surface at 0° incidence angle.

**Figure 22 polymers-14-05444-f022:**
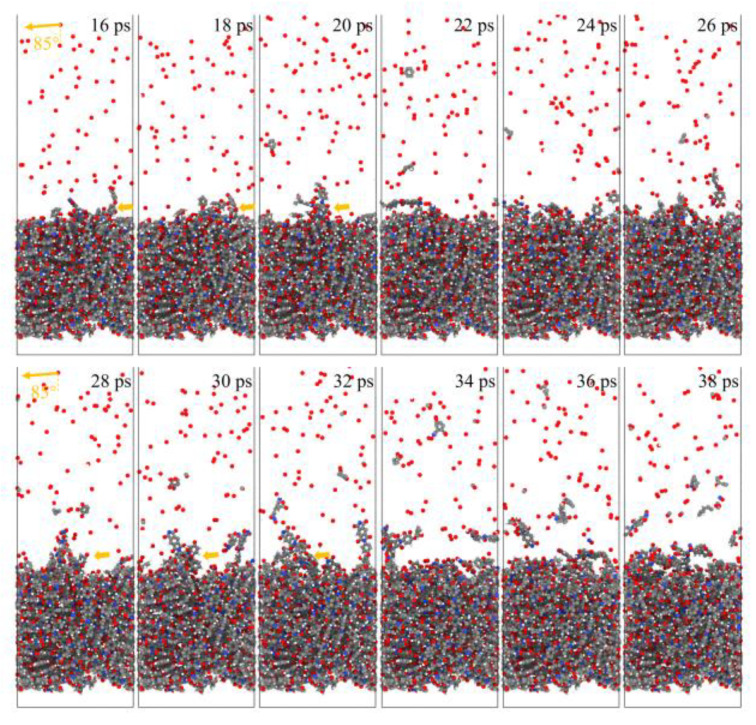
Evolution of the initial phase of atomic oxygen impacting Kapton material surface at 85° incidence angle.

**Figure 23 polymers-14-05444-f023:**
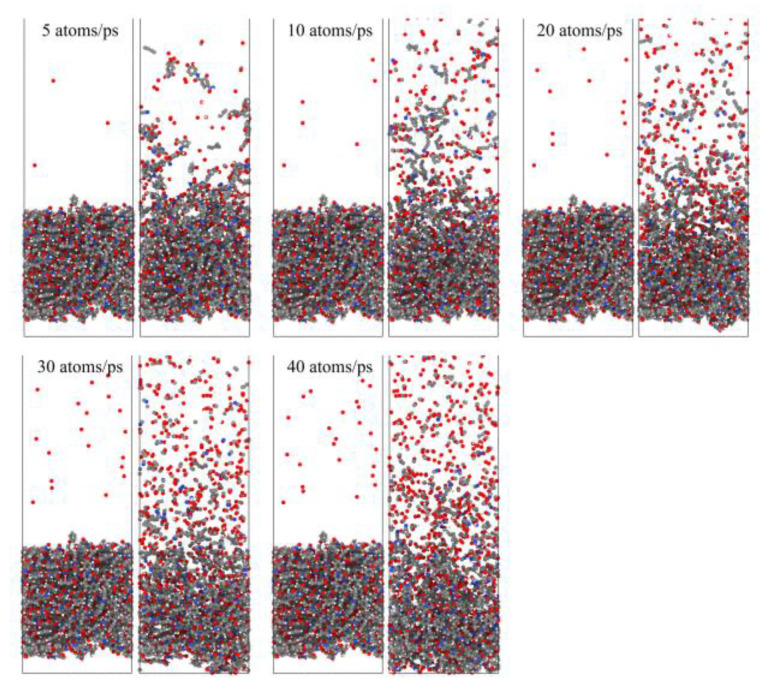
Snapshots of AO impacting Kapton at different dose rates during the initial phase (0.6 ps) and the termination phase (30 ps).

**Figure 24 polymers-14-05444-f024:**
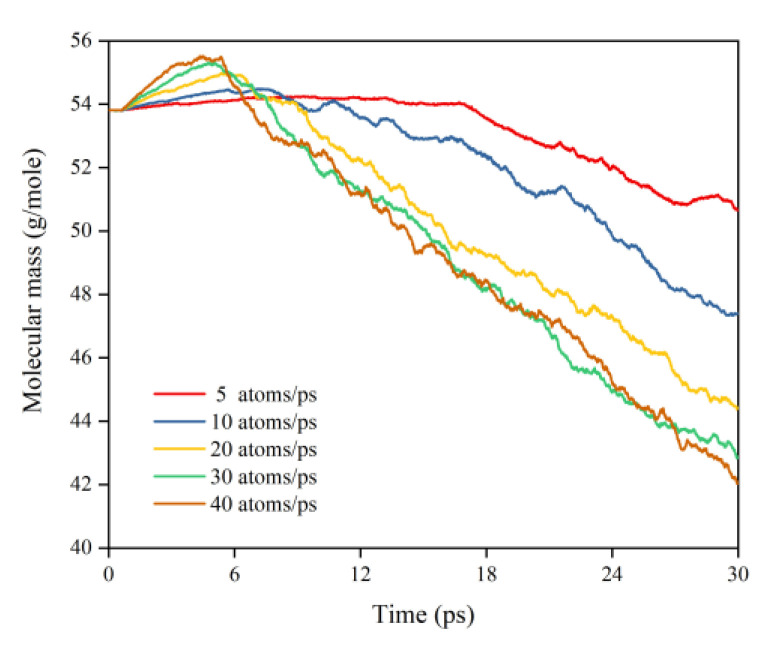
Mass changes of the Kapton during atomic oxygen impact at different dose rates.

**Figure 25 polymers-14-05444-f025:**
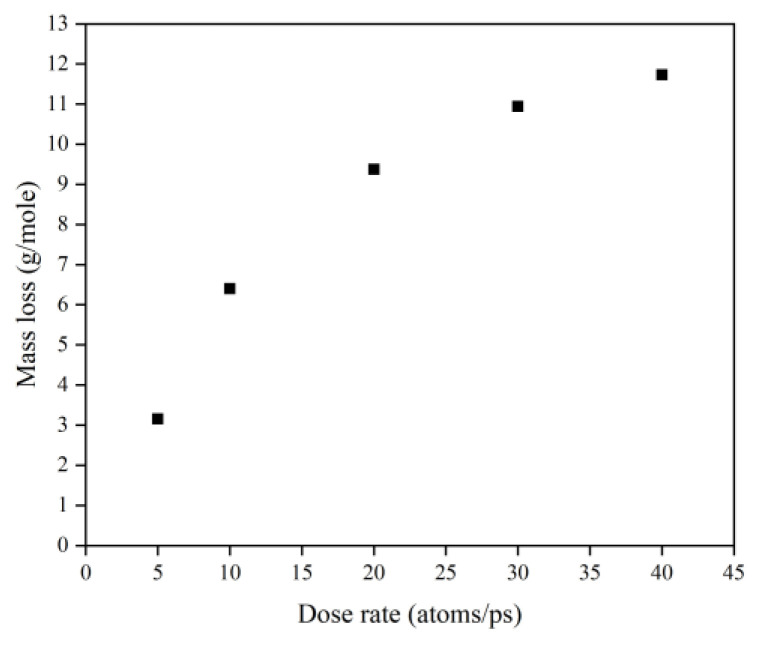
Mass loss of the Kapton during atomic oxygen impact at different dose rates.

**Figure 26 polymers-14-05444-f026:**
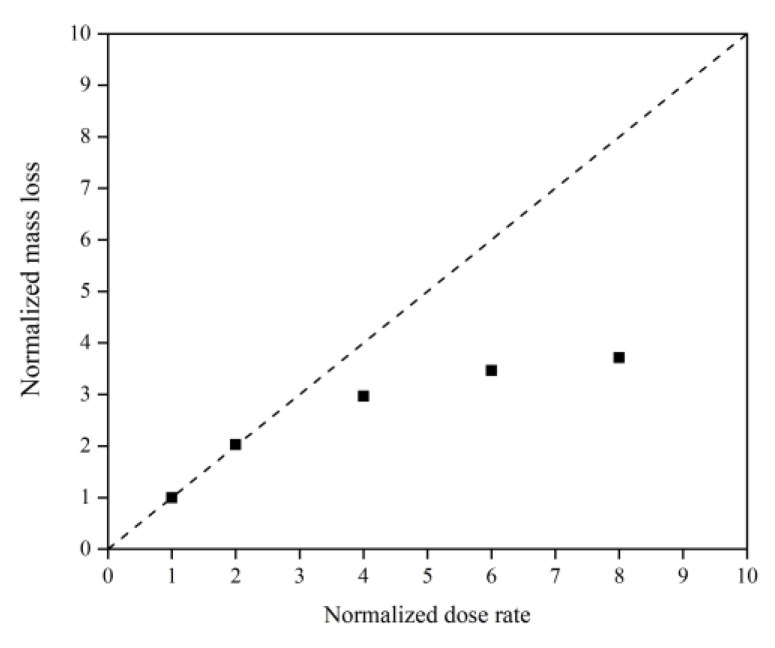
Normalized mass loss of the Kapton during atomic oxygen impact at different dose rates.

**Figure 27 polymers-14-05444-f027:**
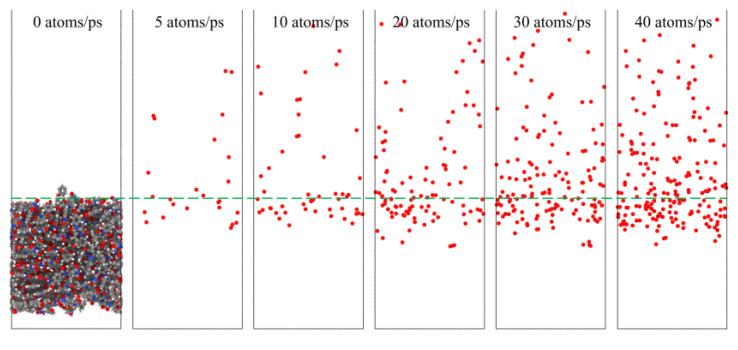
Snapshots of atomic oxygen trajectories at different dose rates in the initial phase (5 ps).

**Figure 28 polymers-14-05444-f028:**
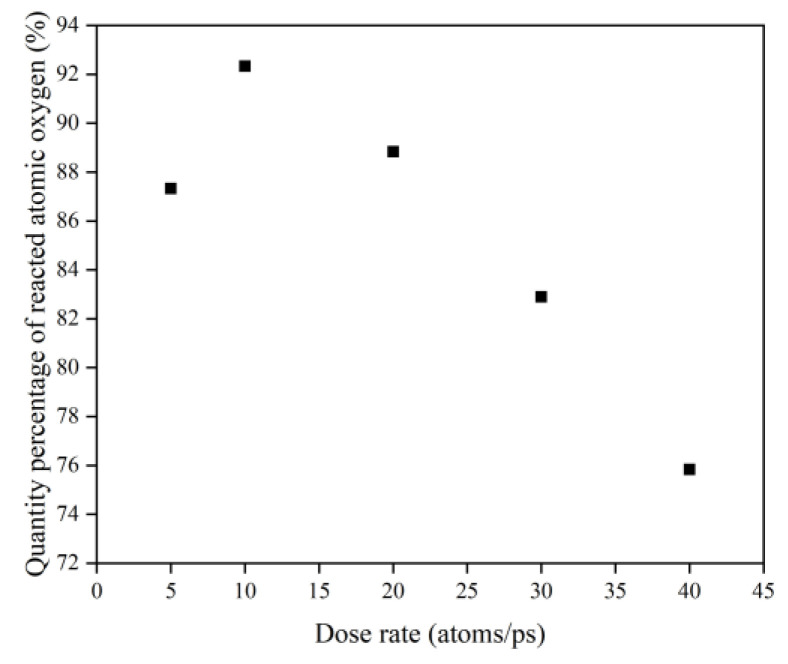
Quantity percentage of reacted atomic oxygen.

**Figure 29 polymers-14-05444-f029:**
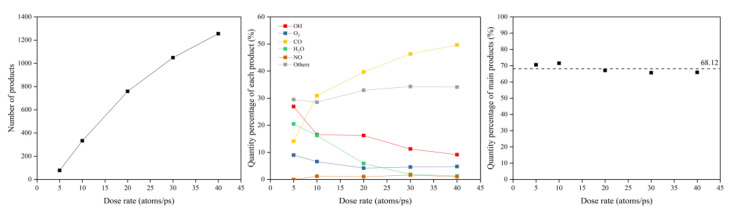
Product information of the Kapton after atomic oxygen impact at different dose rates.

**Figure 30 polymers-14-05444-f030:**
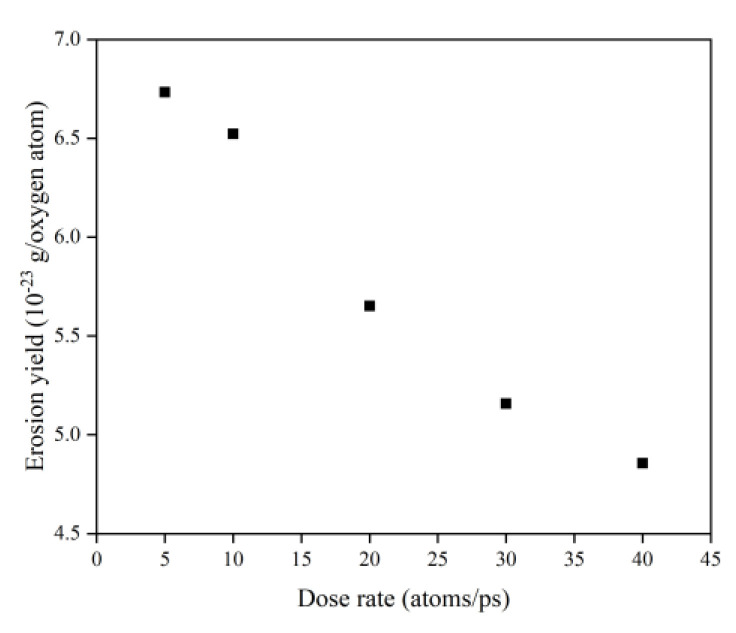
Erosion yield of the Kapton under atomic oxygen impact at different dose rates.

**Figure 31 polymers-14-05444-f031:**
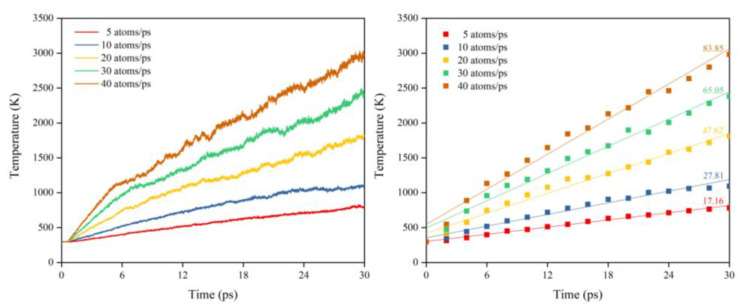
Temperature variation of the Kapton under atomic oxygen impact with different dose rates.

**Figure 32 polymers-14-05444-f032:**
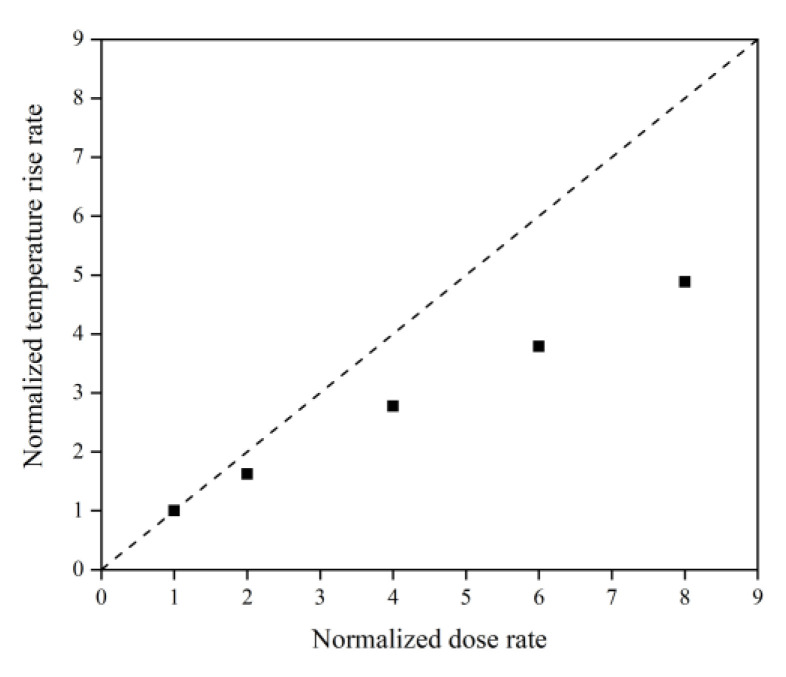
Normalized temperature rise rate of the Kapton under atomic oxygen impact with different dose rates.

## Data Availability

Not applicable.

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
