# Peer review of "Dependence of Incidence Angle and Flux Density in the Damage Effect of Atomic Oxygen on Kapton Film"

_polymers, 2022, doi:10.3390/polym14245444_

Round 1
Reviewer 1 Report
This article studies the dependence of incidence angle and flux density of atomic oxygen on Kapton film. However, the following inquiries should be solved before it is published.
1) Lines 190,363,366 "figure" twice typo repetitively appear in the manuscript.
2) Is 1 ns NPT enough for the full equilibration of the system? How to prove that?
3) Are the 140 PI monomers connected end to end? The morphology snapshot does not show a long chain of the polymer but shows many broken chains.
4) How was the simulation length of 30 ps selected? The reference paper published in 1989 is not comparable with this study.
Reviewer 2 Report
This work described the dependence of incidence angle and flux density in the damaging effect of atomic oxygen on Kapton film. The authors investigated the mass loss and atomic oxygen effect. The different sections of the incidence angle ranges were interesting. I recommend a minor revision on the following points:
1. The connection between the mass loss and atomic oxygen should be explained in a more detailed and clearer way. It seemed that the dose rate and the temperature were all factors. Could the authors build an overall function to describe this connection?
2. The authors claimed that “changes in the angle of incidence and dose rate have little effect on the total content of the main products of the AO effect, averaging around 70%”. Could the authors suggest a method to reduce the AO effect?
3. The language use should be improved. For example, the tense should be checked.
Reviewer 3 Report
In general, the article is scientific. It investigated the effects of incidence angle and flux density on the damaging of AO on LEO. And this article explained the effects by experiments and simulations, in which the simulations methods is different from the conventional methods. Although there are some details should be modified, the main part is logical and substantial. Overall, I would suggest the publication of this manuscript.
Minor comments and suggestions that the authors may want to address.
Comment 1: In the abstract, I have understanded the effect of AO on the LEO environment. However, there are many issues influence the Kapton films and why do you investigate the influence of incidence angle and flux density should be declared.
Comment 2: In the last paragraph of introduction, the results of the investigation should be stated, and the innovative conclusions should be declared by specific data.
Comment 3: For better understanding, recommend adding Figure for describing the experiment setup.
Comment 4: In line 190, the word “Figure” is repeated, please check it and the full text for similar errors, such as line 245.
Comment 5: For Figure 3., 4. and so on, the placement of the picture needs to be reconsidered, the figures are too small to understand the content.
Comment 6: In Figure 9., the cosθ is 1.00 when θ=0, however, in this picture, cosθ is smaller than 0.30 when θ=0, please explain the reasons.
Comment 7: There are many figures lacking the labels, please check and add them, for example Figure 9., 10.
Comment 8: In the conclusion, the second point mentions that there is a critical value for the effect of flux density on the AO effect. Could the critical value be determined or within a range? It’s better for explaining the conclusion.

Round 2
Reviewer 1 Report
Agree to publish in present forms